# Multi-Objective Optimization of the Planting Industry in Jiangsu Province and Analysis of Its "Water-Energy-Carbon" Characteristics

**Yizhen Jia * and Xiaodong Yan**

Business School, Hohai University, Nanjing 211100, China; yanxiaodong@hhu.edu.cn
* Correspondence: jiayizhen027@126.com

**Abstract:** The modern development of the planting industry needs to not only ensure food supply but also to consider social and environmental issues. This poses higher demands for rational planning of planting structures to achieve green development while meeting demands and conserving resources. Therefore, this paper takes Jiangsu Province as a representative case, accounting for and analyzing the water footprint, energy consumption, and carbon emissions of seven major crops. Based on this analysis, a multi-objective planning model is established to explore the optimization of its planting structure. The results show that: (1) from 2010 to 2020, the overall water footprint of these seven crops in Jiangsu Province initially increased and then decreased, while energy consumption showed a fluctuating upward trend; (2) carbon emissions from planting in Jiangsu Province initially increased and then fluctuated downward over time, and exhibited significant spatial clustering characteristics, with overall emissions being higher in northern Jiangsu, followed by central Jiangsu, and then southern Jiangsu; (3) the optimization results indicate that economic benefits increased by 4.06%, while the carbon emission and grey water footprints decreased by 3.78% and 7.62%, respectively, resulting in comprehensive improvements in economic and ecological benefits. This study can provide theoretical support for adjusting the planting structure in crop-producing areas and promoting the green and sustainable development of the planting industry.

**Keywords:** crop planting; feature analysis; planting structure; multi-objective optimization; green sustainability



## 1. Introduction

Crop production is one of the oldest activities in human society and serves as a fundamental guarantee for survival and social stability [1]. Among the 17 Sustainable Development Goals (SDGs) announced by the United Nations in 2015, SDG2 focuses on eliminating hunger and achieving food security [2,3]. China, with only 9% of the world's land, feeds 20% of the global population, and its grain production has remained relatively stable in recent years [4]. However, it is important to note that challenges such as exacerbated climate change, agricultural environmental pollution, and uneven distribution of resources persist [5]. According to existing research, planting structure significantly influences food security and rational resource utilization [6,7]. Thus, it is necessary to clarify the current status of resource consumption and pollution in key crop production areas and explore paths for optimizing crop planting structures.

Water resources and energy consumption are crucial input factors in crop production. Water resources are used for crop irrigation and dilution of fertilizers and pesticides [8], while energy is utilized for mechanized farming and the manufacturing and usage processes of agricultural materials [9]. Additionally, carbon emissions generated during crop production have garnered attention from various sectors of society [10]. According to studies, agricultural greenhouse gas emissions in China account for approximately 17% of the total greenhouse gas emissions [11,12]. The "2023 China Agricultural and Rural

Low-carbon Development Report" points out that agricultural carbon emissions are fundamental and survival-related emissions, and efforts should be made to promote high yields, efficiency, and green low-carbon development. As resources become increasingly scarce in the field of planting, low-carbon and green production has become an inevitable requirement for food security and sustainable development. Incorporating crop carbon emissions into the decision-making and management processes of planting industries holds significant practical significance.

To ensure food security and achieve sustainable development in the planting industry, scholars have reviewed the current status and issues of food security [13,14]. They have conducted research on various aspects, including the construction and measurement of food security indicators [15], farmland protection and demand forecasting [16], technologies for increasing grain production [17–19], and factors affecting agricultural productivity and food security [20–22]. Based on these studies, researchers have focused their attention on the collaborative examination of planting industries and other resources, as crop production processes are influenced by climate and environment and involve multiple input factors. Guo et al. [23] proposed four agricultural nitrogen use improvement strategies that can increase grain yield and improve environmental issues. Shi et al. [24] evaluated the efficiency of agricultural water resource utilization in the Yangtze River Economic Belt and explored spatial network characteristics using social network analysis methods. These studies focused on the utilization and protection of relevant resources, providing a theoretical basis for the formulation of planting industry production policies and planning.

Research on carbon emissions and low-carbon production in the planting industry, both domestically and internationally, continues to advance. Traditional research topics include definition and scope analysis [25,26], emission estimation [27], analysis of associated factors and spatial effects [28], and discussions on decarbonization methods and management [29–31]. With the development of the internet and innovative technologies, innovative research fields such as green Internet of Things agriculture [32,33] and innovative urban agriculture [34] have emerged gradually. Scholars from various perspectives have organized and analyzed the current situation, proposing various investigation schemes and implementation measures and striving to resolve the contradiction between agricultural stability, increased supply, and green production. In recent years, Chinese scholars have focused on analyzing the spatial effects [35], emission reduction potential [36], and peak trends [37] of agricultural carbon emissions based on traditional calculations.

From the perspective of research methods, the field of carbon emissions and green production in the planting industry involves various qualitative and quantitative methods. Based on the extended Theory of Planned Behavior, Li et al. [38] empirically analyzed the willingness and behavioral factors of farmers adopting a rice-green manure rotation system. Liu et al. [39] used the Super-SBM model to calculate China's agricultural green total factor productivity based on carbon emissions. He et al. [40] used a random effects panel Tobit model to examine the role of the adoption rate of agricultural green production technologies in enhancing low-carbon efficiency. Among these, multi-objective optimization methods control the optimization direction through goal setting and restrict resource inputs through constraint conditions, which is beneficial for the comprehensive improvement of crop production processes. Gong et al. [41] established a non-precise interval programming model to obtain optimal irrigation planting structure schemes under different hydrological year conditions. Li et al. [42] established an optimization model for sustainable management of agricultural water, food, and energy relationships under uncertain conditions. In terms of research areas, optimization of agricultural planting structures involves the national [43,44], regional [45,46], and provincial or municipal levels [47–49].

Based on the above review of previous research, scholars have shown a high level of concern for production management and resource utilization in the planting industry. Although significant progress has been made in optimizing the planting structure, there are gaps remaining to be bridged. First, existing studies on optimization of planting structures have mostly focused on water resource management, with investigations into pollution

being limited to the use of single materials such as fertilizers and pesticides and research that incorporates carbon emissions into optimization goals being lacking. Second, the grey water footprint, which reflects the ecological negative externality of crop water use, has become a research direction in recent years regarding farmland pollution [50]. However, there is limited research considering the grey water footprint in the optimization process of planting structures. Jiangsu Province, as an important planting base in China, plays a leading role in the construction of modern agriculture [51]. However, it faces challenges such as significant pressure on arable land, limited per capita water resources, and increasing inputs of materials such as fertilizers, leading to issues such as farmland pollution which constrain the development of the planting industry. Taking Jiangsu Province as a representative example, it is necessary, to incorporate the minimization of the carbon emission and gray water footprints into the objective function. Research on crop planting structure optimization paths and comprehensive management measures should be conducted based on this foundation.

Therefore, in light of the increasingly prominent issues of carbon emissions and water pollution in the planting industry, this paper considers the important resource elements in the planting system and deepens existing research from the following three aspects. First, it utilizes mainstream calculation methods to calculate the water footprint, energy consumption, and carbon emissions of seven crops in Jiangsu Province, striving for scientific accuracy in the calculations. Second, it examines the characteristics of changes in various elements and the differences between crops while analyzing possible reasons for these differences. Finally, based on the above analysis and aiming to maximize economic benefits and minimizing the carbon emission and grey water footprints, it explores optimized planting structure schemes. This study aims to provide theoretical support for adjusting the planting structure in crop-producing areas and promoting the green and sustainable development of the planting industries.

The rest of this paper is organized as follows. Section 2 provides an overview of the research area, describing the calculation methods for water footprint, energy consumption, and carbon emissions, introducing the multi-objective planning model for planting structure constructed, and outlining the data sources. Section 3 analyzes the water footprint, energy consumption, and temporal–spatial variation characteristics of the carbon emissions of crops in Jiangsu Province, as well as the results of the optimized planting structure. Finally, Sections 4 and 5 discuss the rationality of the research results, the feasibility of the optimization scheme in production practice, and the resulting policy implications.

## 2. Materials and Methods

### 2.1. Research Area Overview and Research Approach

Jiangsu Province is located in the eastern coastal region of China downstream of the Yangtze and Huaihe rivers, and has a total area of $1.07 * 10^5$ km$^2$. As of October 2020, the permanent population was 84.748 million. The land resources are predominantly plains, accounting for 86.9% of the total area, characterized by deep and fertile soil. The province experiences an East Asian monsoon climate, with average annual precipitation of about 704–1250 mm in various regions.

Jiangsu is a major economic province in China and is one of the country's 13 major grain-producing provinces. However, its arable land area accounts for only 4.6% of the national total, with a per capita arable land area of about 0.048 hm$^2$, reflecting the regional characteristic of high population density and limited land resources. In 2020, Jiangsu's grain production ranking dropped from fifth in 2011 to seventh nationally, mainly due to limited arable land. The challenge for the future is how to tap into production potential and engage in scientific planning on this limited arable land area.

Agriculture is a significant consumer of water resources, with Jiangsu's total water supply reaching $4.53 * 10^{10}$ m$^3$ in 2020, of which agricultural water usage accounted for $2.67 * 10^{10}$ m$^3$, or 58.9%. Additionally, the province aims to promote agricultural modernization, focusing on the development of agricultural science, technology, and

equipment. As a result, the level of agricultural mechanization is increasing. However, this has led to further increases in agricultural inputs. In 2020, fertilizer application in Jiangsu Province reached 2.875 million tons, accounting for 5.35% of the national total [52]. The corresponding rise in energy consumption also contributes to an increase in agricultural carbon emissions to an extent.

The "Carbon Peaking Implementation Plan for Urban and Rural Construction in Jiangsu Province" released in 2023 requires the integration of the concept of green and low-carbon development into various aspects of urban and rural construction. This poses additional challenges for the task of building green villages in the new era.

This article's model for optimizing agricultural planting structure is based on an accounting of the water footprint, energy consumption, and carbon emissions in crop production. Using multi-factor constraints as the starting point, the model focuses on selecting economic and environmental indicators as targets. According to the research results, this paper analyzes the composition of the water footprint and energy consumption for major crops in Jiangsu Province. It explores the spatiotemporal characteristics of agricultural carbon emissions and deduces the optimized planting structure for agriculture; the research technology roadmap is illustrated in Figure 1.

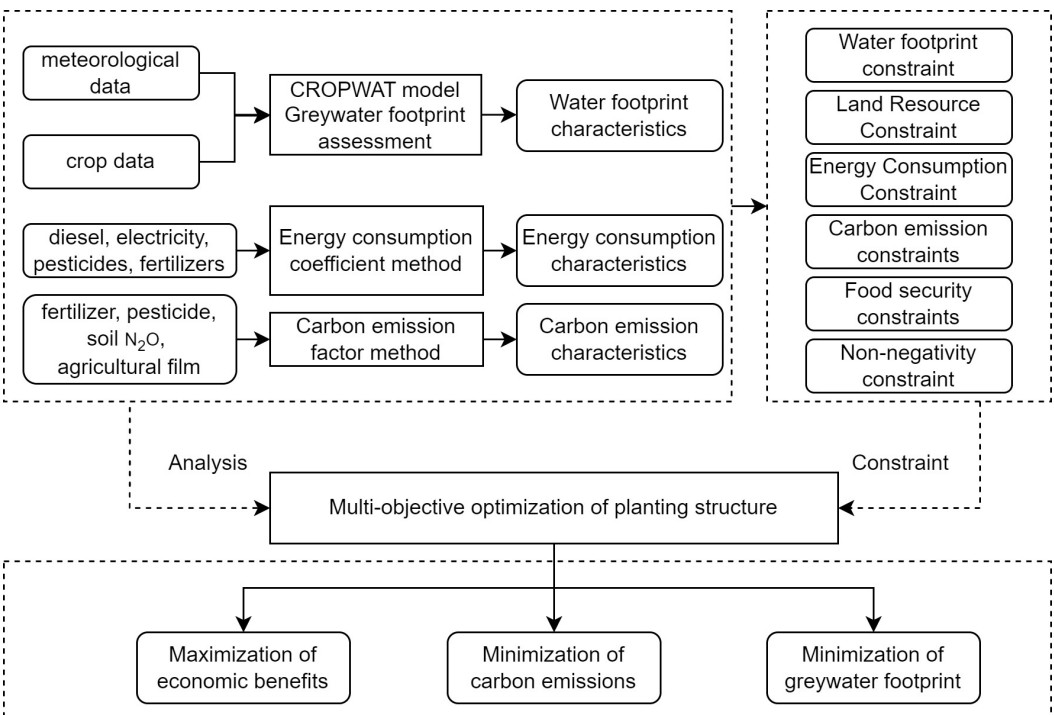

**Figure 1.** The overall research framework.

### 2.2. Water Footprint Accounting

The total water footprint of crops ($W$) is the sum of the blue water footprint, green water footprint, and grey water footprint [53,54], which respectively refer to the surface water and groundwater used for crop irrigation, the effective precipitation absorbed and utilized by crop growth, and the water consumed for diluting pollutants in farmland.

$$W = W_{blue} + W_{green} + W_{grey} \tag{1}$$

The blue water and green water footprints are typically calculated using the CROPWAT model proposed by the Food and Agriculture Organization of the United Nations (FAO) [55]. This method assumes that crop growth conditions are in the optimal state,

where the crop's water requirement equals the evapotranspiration amount. The blue water and green water footprints of crops are then calculated using the following formulas [56]:

$$W_{blue} = 10 * \sum_{d=1}^{n} max(0, \sum_{d=1}^{n} (K_c * ET_0) - P_{eff}) \tag{2}$$

$$W_{green} = 10 * \sum_{d=1}^{n} min(\sum_{d=1}^{n} (K_c * ET_0), P_{eff}) \tag{3}$$

$$P_{eff} = \begin{cases} P(\frac{125 - 0.6 * P}{125}), P \leq \frac{250}{3} \\ \frac{125}{3} + 0.1 * P, P > \frac{250}{3} \end{cases} \tag{4}$$

where $W_{blue}$ and $W_{green}$ respectively represent the crop's unit area blue water footprint ($m^3/hm^2$) and green water footprint ($m^3/hm^2$), 10 is the unit conversion factor, $P_{eff}$ represents the ten-day total precipitation during the crop's growing season (mm), with the calculation process completed in "CROPWAT 8.0" software, $K_c$ is the crop coefficient determined by referencing the recommended crop data database and considering the actual conditions of the region, $ET_0$ represents the reference crop evapotranspiration (mm/d), obtained from the calculation using Penman–Monteith formula, and $P$ represents the ten-day total precipitation (mm).

The calculation of the crop's unit area grey water footprint refers to the improved method proposed by Chu, Huang, Lai, Yang, and Hou [57], which takes into account the natural environment's ability to degrade pollutants:

$$W_{grey} = \frac{\mu * AR * (1 - \alpha)}{C_{max} - C_0} \tag{5}$$

where $W_{grey}$ represents the crop's unit area grey water footprint ($m^3/hm^2$), $\mu$ denotes the leaching rate indicating the proportion of the quantity of substances entering the water body causing pollution to the total amount of chemical substances applied (the leaching rate for nitrogen fertilizer is 10% [58]), $AR$ represents the amount of nitrogen fertilizer applied per unit area of the crop ($kg/hm^2$), $\alpha$ represents the degradation coefficient of water pollution by channels (for the middle and lower reaches of the Yangtze River region, the degradation coefficient is 21.9 [57]), $C_{max}$ represents the maximum concentration of pollutants that the environment can accommodate, and $C_0$ represents the natural background concentration of nitrogen elements in the water body, typically assumed to be 0.

### 2.3. Energy Consumption and Carbon Emissions Accounting

This study calculates the energy consumption and carbon emissions of various crops using the energy consumption coefficient method and emission factor method. This involves multiplying the relevant coefficients by the quantities of various elements needed per unit area of the crop. The calculation in this paper focuses on the energy consumption per unit area of various crops, including consumption of diesel, electricity, pesticides, and fertilizers [59]:

$$E = \frac{f_1 + f_2 * M}{P_1} * N_1 + \frac{f_3 - f_4}{P_2} * N_2 + \frac{f_5}{P_3} * N_3 + \sum_{i=1}^{4} EI_{4,i} * N_{4,i} \tag{6}$$

where $E$ is the unit area energy consumption of crops ($GJ/hm^2$), $N_1$, $N_2$, and $N_3$ are the energy consumption coefficients for diesel, electricity, and pesticides, respectively, $N_{4,i}(i = 1,2,3,4)$ represents the energy consumption coefficients for the four fertilizers (nitrogen, phosphorus, potassium, and compound fertilizer), $EI_{4,i}(i = 1,2,3,4)$ represents the unit area application rate of fertilizer $i$ in kg, $P_1$, $P_2$, and $P_3$ are the respective prices of diesel, electricity, and pesticides ($CNY/kg$), $f_1, f_2, f_3, f_4, f_5$ respectively represent the unit area fuel power cost, machinery operation cost, irrigation and drainage cost, water

cost, and pesticide cost (*CNY*), and *M* is the proportion of fuel power cost in machinery operation cost, assumed to be 40% [60].

Carbon emissions include carbon emissions from agricultural inputs and soil $N_2O$ emissions. Carbon emissions from fertilizers, pesticides, and agricultural films account for about 80% of the total carbon emissions from crops [61]. Therefore, this study selects these three elements for calculation:

$$C = \sum_{j=1}^{3} T_j * \sigma_j + L * \delta \tag{7}$$

where *C* is the unit area carbon emissions of crops (kg C/hm$^2$), $T_j(j = 1, 2, 3)$ represents the usage of fertilizers, pesticides, and agricultural films (the calculation method for pesticide usage is the same as above, and fertilizer usage is the sum of the four types of fertilizers), $\sigma_j(j = 1, 2, 3)$ represent the carbon emission coefficients for fertilizers, pesticides, and agricultural films, respectively, which are 0.8956 kg C/kg, 4.9341 kg C/kg, and 5.18 kg C/kg [62], *L* represents the soil carbon $N_2O$ emission coefficient for each crop (kg/hm$^2$), obtained from the reference literature [63], and $\delta$ represents the coefficient for converting $N_2O$ to *C*, which is 81.2727.

### 2.4. Establishment of the Crop Planting Structure Optimization Model

The planting structure is influenced by local natural conditions, societal perspectives, economic development, etc. Jiangsu Province has strong grain production capacity; the challenge lies in how to tap into the potential for cultivating multiple crop varieties and achieve diverse objectives. This study, based on major sowing crop data in Jiangsu Province, takes the area of seven crops (rice, wheat, corn, legumes, potatoes, oilseeds, and vegetables) $X_j(j = 1, 2, ..., 7)$ as decision variables. It then incorporates the key tasks of pollution reduction and carbon reduction into the objective system, sets goals on economic and environmental levels, and constructs an optimization model for the agricultural planting structure in Jiangsu Province.

#### 2.4.1. Objective Function

(1)  Maximization of economic benefits

According to the "National Compilation of Agricultural Product Cost and Benefit", the main product output value is used to measure agricultural income, while the cash cost index is used as the cost of crop cultivation. This study takes the difference between the main product output value and the cash cost as the parameter for the economic benefit indicator. The objective function for maximizing economic benefits is as follows:

$$max\ f_1(X) = \sum_{j=1}^{7} (G_j - H_j) X_j \tag{8}$$

where $f_1(X)$ is the total economic benefit (CNY), $G_j$ is the unit area main product output value of the *j*-th crop (CNY/hm$^2$), $H_j$ is the unit area cash cost of the j-th crop (CNY/hm$^2$), and $X_j$ is the planting area of the j-th crop (hm$^2$).

(2)  Minimization of carbon emissions

In recent years, environmental issues caused by carbon emissions have become increasingly urgent. China is putting forward the "dual carbon" goals, and Jiangsu Province's low-carbon development is imposing new requirements on the agricultural sector. Therefore, this paper sets the objective function for minimizing carbon emissions as follows:

$$min\ f_2(X) = \sum_{j=1}^{7} C_j X_j \tag{9}$$

where $f_2(X)$ is the total carbon emissions (kg C) and $C_j$ is the unit area carbon emissions of the $j$-th crop (kg C/hm$^2$).

(3) Minimization of grey water footprint

It is essential to reduce water resource consumption in order to dissolve and dilute pollutants. The saved water can be utilized in various production and daily life areas. The objective function for minimizing the grey water footprint is as follows:

$$min \ f_3(X) = \sum_{j=1}^{7} W_{grey,j} X_j \tag{10}$$

where $f_3(X)$ is the total grey water footprint (m$^3$) and $W_{grey,j}$ is the unit area grey water footprint of the $j$-th crop (m$^3$/hm$^2$).

2.4.2. Constraint Conditions

Taking into account the consumption status of associated elements in the planting system, this paper establishes constraint conditions for the multi-objective optimization model of agricultural planting structure in Jiangsu Province. These constraints include water footprint constraints, land resource constraints, energy consumption constraints, carbon emissions constraints, food security constraints, and non-negativity constraints.

(1) Water footprint constraint

The water footprint reflects the amount of water resources required during the growth process of crop planting. The optimized total water footprint should not exceed the total water footprint in 2020, denoted as $W^0$. The water footprint per unit area for each crop is represented by $W_j$.

$$\sum_{j=1}^{7} W_j X_j \leq W^0 \tag{11}$$

(2) Land Resource Constraint

The land resource constraint primarily considers the total arable land area and the arable land area for grain crops. The area cultivated for crops should be less than the available arable land area, which is determined by the existing arable land area $S_A$ and the replanting index $a$ [64]. The existing arable land area in this study uses the results from "Main Data Bulletin of the Third National Land Survey in Jiangsu Province", while the replanting index is calculated from statistical data.

$$\sum_{j=1}^{7} X_j / a \leq S_A \tag{12}$$

According to the "Jiangsu Province 14th Five-Year Plan for Comprehensive Promotion of Rural Revitalization and Acceleration of Agricultural and Rural Modernization" (hereinafter referred to as the "Plan"), by 2025 the sown area of grain crops ($S_{A,f}$) should be stable at over $8 * 10^7$ mu, the rice planting area should be stable at $3.2 * 10^7$ mu, and the sown area of vegetables should be stable at over $2 * 10^7$ mu.

$$\sum_{j=1}^{5} X_j / a \geq S_{A,f} \tag{13}$$

(3) Energy Consumption Constraint

Crop energy consumption reflects the various energy inputs during the cultivation and irrigation process of crops. The optimized total energy consumption should not exceed

the total energy consumption in 2020, denoted as $E^0$. The energy consumption per unit area for each crop is represented by $E_j$.

$$\sum_{j=1}^{7} E_j X_j \leq E^0 \tag{14}$$

(4)    Carbon emission constraints

In order to effectively reduce carbon emissions in agricultural crops, this study establishes a carbon emission minimization objective function. Simultaneously, the optimized total carbon emissions are required to be no greater than the total carbon emissions in the year 2020, denoted as $C^0$. The carbon emissions per unit area for each crop are represented by $C_j$.

$$\sum_{j=1}^{7} C_j X_j \leq C^0 \tag{15}$$

(5)    Food security constraints

According to the contents of the plan, the grain total production target for Jiangsu Province by 2025 ($P_f$) is set at over $3.7 * 10^7$ t. This paper involves grain crops, which account for approximately 98% of the total grain crops. Therefore, the total production target is multiplied by the coefficient representing the proportion of the research crop's yield, denoted as $\lambda$. The per unit area yield of the crop is represented by $Y_j$. The vegetable production target ($P_v$) is maintained at over $5.5 * 10^7$ t.

$$\sum_{j=1}^{5} Y_j X_j \leq \lambda P_f \tag{16}$$

$$Y_7 X_7 \geq P_v \tag{17}$$

(6)    Non-negativity constraint

$$X_j \geq 0, j = 1, 2, ..., 7 \tag{18}$$

*2.5. Data Source*

This study constructed a multi-objective optimization model for the planting structure of seven crops in Jiangsu Province. Data on the historical planting area and yield per unit area of crops were obtained from the "Jiangsu Statistical Yearbook" and the "China Rural Statistical Yearbook". The meteorological data used in the calculation of crop blue and green water footprints were obtained from the National Meteorological Science Data Center and processed to generate the dataset "China Surface Climate Daily Value Data Set V3.0", including precipitation, average wind speed, average relative humidity, sunshine duration, maximum temperature, and minimum temperature. The growth stages, crop coefficients, crop heights, and soil parameters for various crops were adopted from the standard parameters provided by the Food and Agriculture Organization (FAO). These parameters were then adjusted according to the actual conditions in Jiangsu Province. The blue and green water footprints for each crop were calculated by averaging the results from the water footprint calculations in Nanjing, Wuxi, and Xuzhou. Data on the main product output value, cash costs, fertilizer application, plastic film usage, machinery operation costs, fuel and power costs, irrigation and drainage costs, water costs, and pesticide costs per unit area of crops were obtained from the "National Compilation of Agricultural Cost-Benefit Data". The energy consumption coefficients for diesel, electricity, pesticides, and fertilizers were sourced from the "Agricultural Technology and Economic Handbook". Prices for diesel, electricity, pesticides, and fertilizers were obtained from the "China Price Yearbook" and the "China Price Statistical Yearbook".

## 3. Results

### 3.1. Analysis of Water and Energy Consumption Characteristics

3.1.1. Water Footprint Characteristics

The annual variation in the total water footprint of seven crops is illustrated in Figure 2. From 2010 to 2020, the range of total water footprint for agricultural crops was $4.7 * 10^{10}$ m$^3$ to $5.45 * 10^{10}$ m$^3$, with an average of $4.99 * 10^{10}$ m$^3$, showing significant overall volatility. The highest point during the study period was observed in 2013, primarily due to a lower effective precipitation that year, resulting in higher irrigation water demand. Prior to 2013, the total water footprint showed a fluctuating upward trend, indicating a continuous increase in the demand for water resources in crop production. After 2017, there is a year-on-year decline in the total water footprint, with an average annual decrease of 3.26%. This reduction is attributed to the advancement of modern agricultural technologies, leading to more efficient utilization of green water resources by crops and a consequent decrease in irrigation water consumption.

Furthermore, based on Figure 2, an analysis was conducted to examine the proportion and variations of each crop in the agricultural water footprint. Among them, rice and wheat contribute the most to the water footprint, with vegetables ranking third. These three crops collectively account for over 87% of the total water footprint. The total water footprint of rice exhibits a fluctuating downward trend, while that of wheat remains relatively stable. Legumes and potatoes have the smallest water footprints, accounting for 1.58% and 0.52%, respectively, with slight fluctuations and decreases. The water footprint of oilseeds shows a fluctuating downward trend, with an average annual decrease of 6.95%. The water resource requirements and changing trends of crops are related to the suitable growth conditions, climate, and planting scale for each crop.

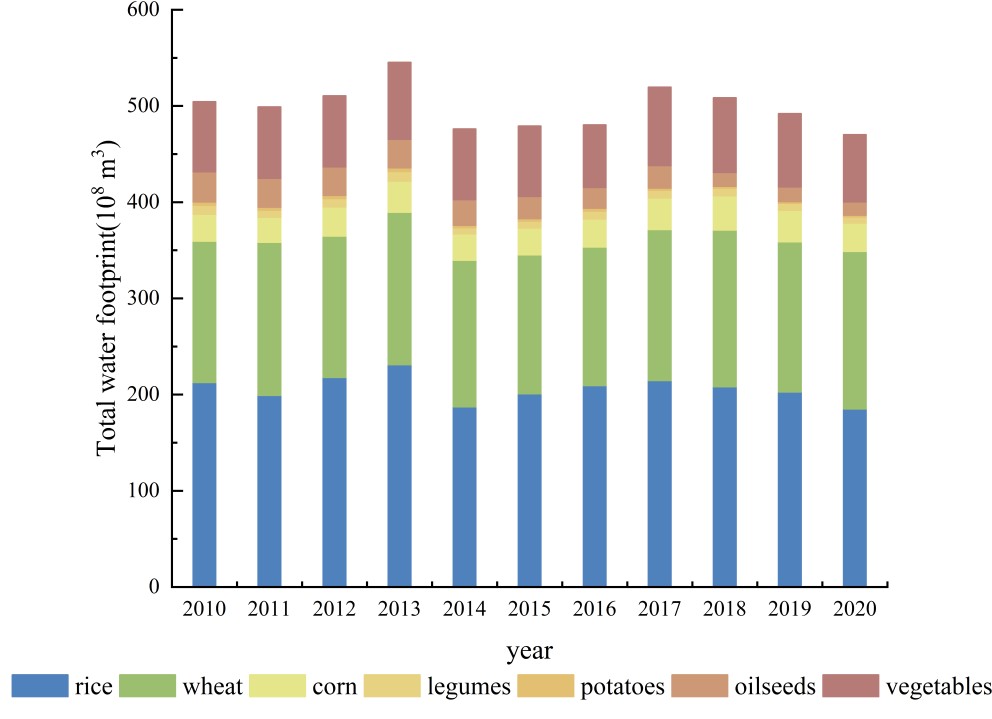

**Figure 2.** Total water footprint of agricultural crops in Jiangsu province from 2010 to 2020.

The structural characteristics of the water footprint of the crops are illustrated in Figure 3. The total water footprint of the crops is primarily composed of blue water and green water, with both averaging close to 40% over the study period and showing relatively small annual variations. Selecting the top three crops with the highest water footprint contribution for analysis, the water footprint structure for rice is generally consistent with the overall water footprint structure. The grey water and green water footprints remain

relatively stable, while the blue water footprint fluctuates significantly, indicating that the irrigation water demand for rice is significantly influenced by yearly variations. In the case of wheat, the blue water proportion is substantial, accounting for 43%, and both the blue water and green water footprints exhibit noticeable annual fluctuations. In the case of vegetables, green water consumption constitutes the largest proportion, indicating high efficiency in rainwater utilization. Compared to the other two grain crops, vegetables have a higher proportion of grey water footprint. The water pollution caused by the application of chemical fertilizer during the vegetables planting process is obvious, showing that attention should be paid to the sewage problem during planting in the future.

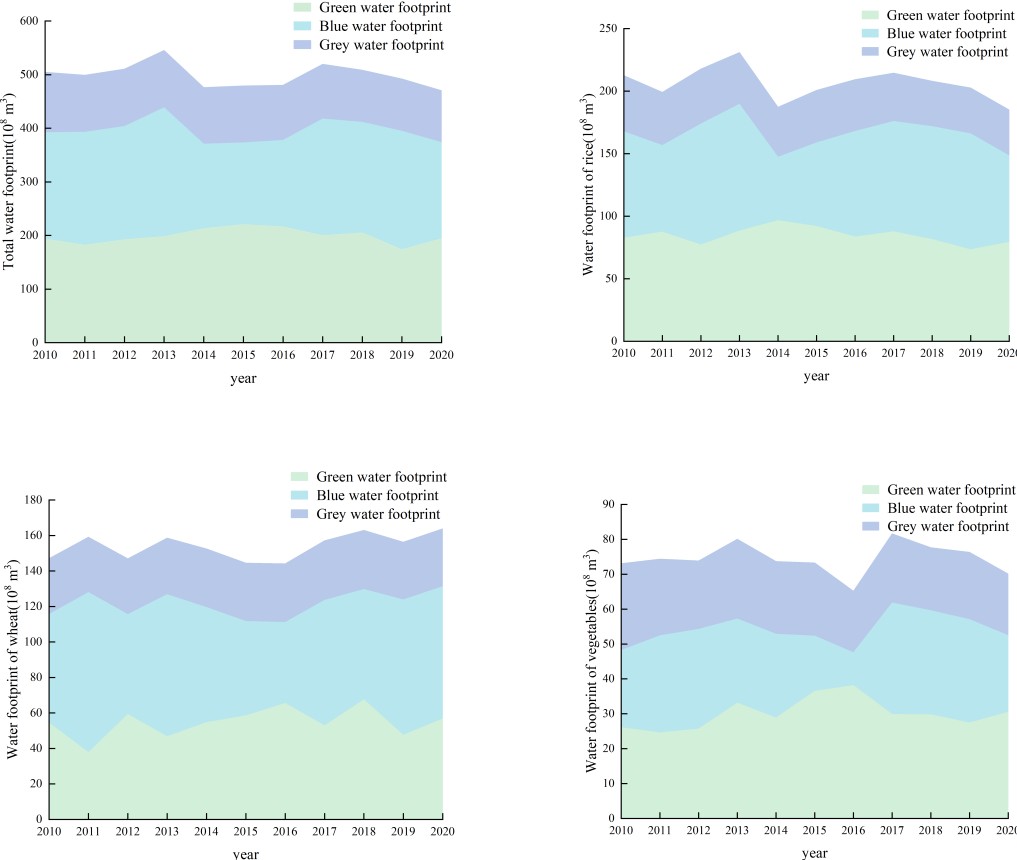

**Figure 3.** Total water footprint structure and water footprint structures of rice, wheat, and vegetables from 2010 to 2020.

The comparative analysis of the average planting area, unit yield water footprint, and unit area water footprint for seven crops from 2010 to 2020 is presented in Figure 4. There are significant differences between the planting areas of major crops, with rice and wheat being the two crops with the largest planting scales. Specifically, rice has the highest unit area water footprint at 9163.47 $m^3$/$hm^2$, while wheat, with a planting area similar to rice, has a significantly smaller unit area water footprint. The unit yield water footprint for wheat is slightly higher than that of rice, standing at 1336.82 $m^3$/t. This is attributed to the higher unit yield of rice, which enjoys economies of scale. Legumes and oilseeds have relatively smaller planting areas but higher unit yield water footprints. It is necessary to explore more advanced technologies in order to improve the unit yield of legumes and oilseeds. Potatoes have the smallest average planting area among the seven crops, and both the unit area water footprint and unit yield water footprint are at a medium to low level. Vegetables have a moderate planting area and unit area water footprint, with the lowest unit yield water footprint among the seven crops. Therefore, stabilizing the vegetables

planting area can help residents to maintain a balanced diet and nutrition while alleviating pressure on the use of agricultural water resources.

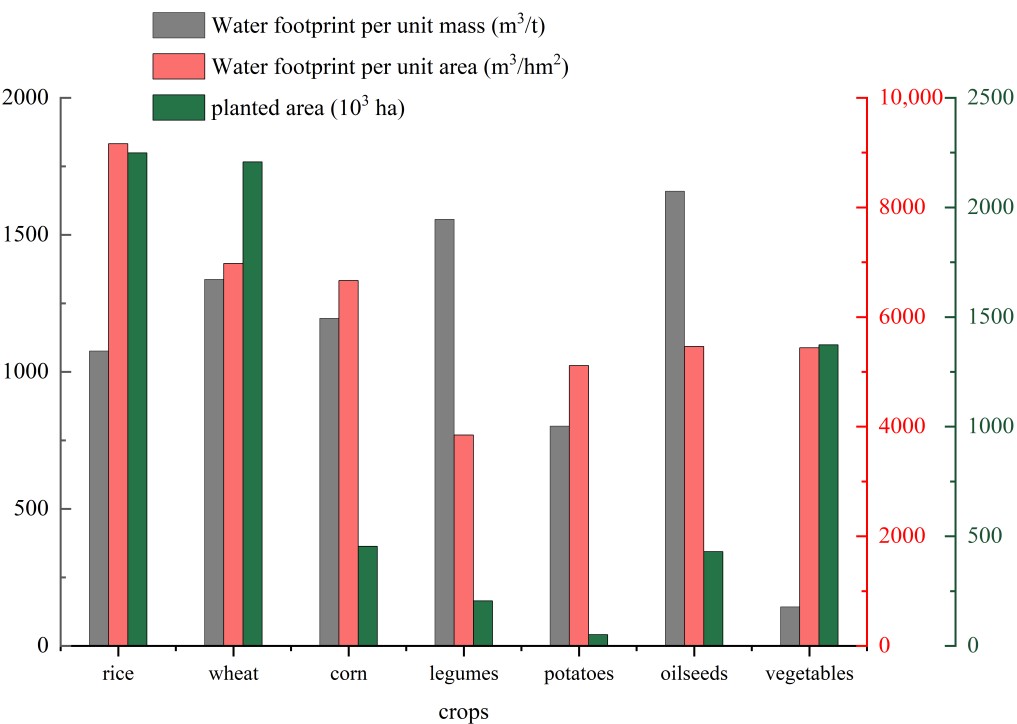

**Figure 4.** Crop planting area, water footprint per unit mass, and water footprint per unit area.

### 3.1.2. Energy Consumption Characteristics

The annual variation in energy consumption for crops shows a fluctuating upward trend, as depicted in Figure 5. The total energy consumption for crops increased by 57.81% from 2010 to 2020. With the development of agricultural modernization, the intensified input of agricultural resources has led to a continuous rise in energy demand. In analyzing the energy consumption for individual crops, rice, with its high water footprint, constitutes a significant portion of crop energy consumption. Vegetables exhibit a relatively high level of energy consumption, primarily attributed to the substantial water resources required in the cultivation process, highlighting the necessity for advanced irrigation techniques. The average energy consumption of wheat is only surpassed by rice and vegetables, and shows a slow upward trend over time. The other crops have relatively lower energy consumption, with legumes and potatoes having the least energy consumption. This is mainly attributed to their smaller planting areas, similar to the water footprint consumption characteristics of these crops. In terms of the overall trend, the total energy consumption for potatoes and oilseeds has slightly decreased, while the energy consumption for other crops has generally fluctuated and increased over the years.

Analyzing the composition of energy consumption, as shown in Table 1, diesel and fertilizer energy consumption account for a relatively high proportion in various crops. Jiangsu Province has a large input of agricultural machinery and fertilizer, which are crucial inputs for increasing crop yield. There are significant differences in the proportion of energy consumption for different types of crops. In the unit area energy consumption of rice and vegetables, electricity accounts for 42.02% and 38.26%, respectively. Both crops have a significant demand for irrigation water, requiring a considerable amount of electricity input during the water pumping and irrigation process. The per unit area diesel consumption for legumes is the highest, reflecting Jiangsu Province's emphasis on the use of and training on legume machinery coupled with the continuous improvement in the level of mechanization in cultivation. Influenced by the growth conditions and planting scale of different crops, the main energy consumption for wheat, corn, potatoes, and oilseeds comes from fertilizer.

The energy consumption of pesticides in various crops is not high, indicating that although pesticides are significant contributors to environmental pollution, their energy consumption in crop cultivation is not prominent.

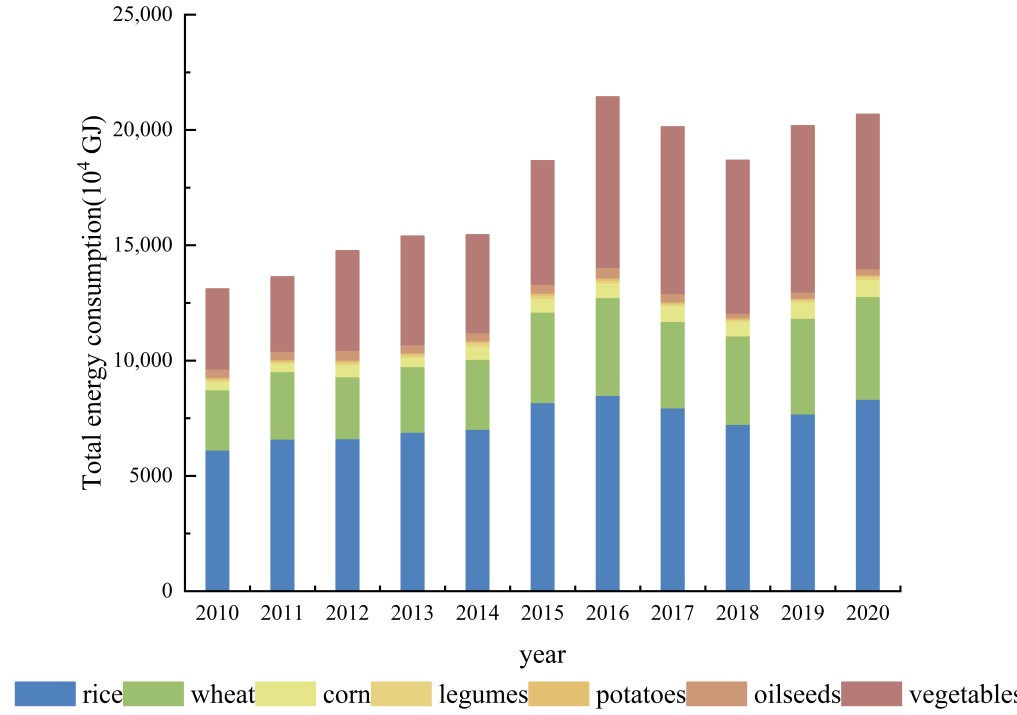

**Figure 5.** Total energy consumption of agricultural crops in Jiangsu province from 2010 to 2020.

**Table 1.** Composition of energy consumption per unit area for crops.

|             | Rice   | Wheat  | Corn   | Legumes | Tuber Crops | Oilseeds | Vegetables |
|-------------|--------|--------|--------|---------|-------------|----------|------------|
| Diesel      | 18.33% | 31.46% | 22.55% | 45.67%  | 25.13%      | 20.96%   | 14.50%     |
| Electricity | 42.02% | 8.49%  | 8.68%  | 17.38%  | 10.91%      | 5.11%    | 38.26%     |
| Pesticides  | 13.69% | 13.18% | 9.07%  | 23.53%  | 13.02%      | 15.57%   | 19.85%     |
| Fertilizer  | 25.96% | 46.87% | 59.70% | 13.42%  | 50.93%      | 58.36%   | 27.38%     |

The unit yield energy consumption for various crops from 2010 to 2020 is shown in Figure 6. The per-unit energy consumption of oil crops consistently ranks the highest, with significant fluctuations over time. It peaked in 2016 due to increased mechanization in oil crop cultivation without a corresponding improvement in per-area yield. In recent years, Jiangsu Province has continuously strengthened the development of oil crop cultivation while appropriately controlling resource inputs. As a result, the per-unit yield of oil crops has increased annually, leading to a decline in per-unit energy consumption followed by a subsequent rebound. Vegetables have the lowest per-unit yield energy consumption, with a mean of 1.03 GJ/t. Comparing the per-unit yields of various crops in 2020, vegetables have a yield of 39.6 t/hm$^2$, with per-unit energy consumption only one-fifth of that of oilseeds. The average per-unit yield energy consumption for the other five crops ranges from 2 to 4 GJ/t, with small fluctuations in corn and potatoes. In 2018, the per-unit yield energy consumption for multiple crops was at a relatively low level and began to rise, mainly due to increased investment in crop cultivation and production as a result of the recent emphasis on agricultural development.

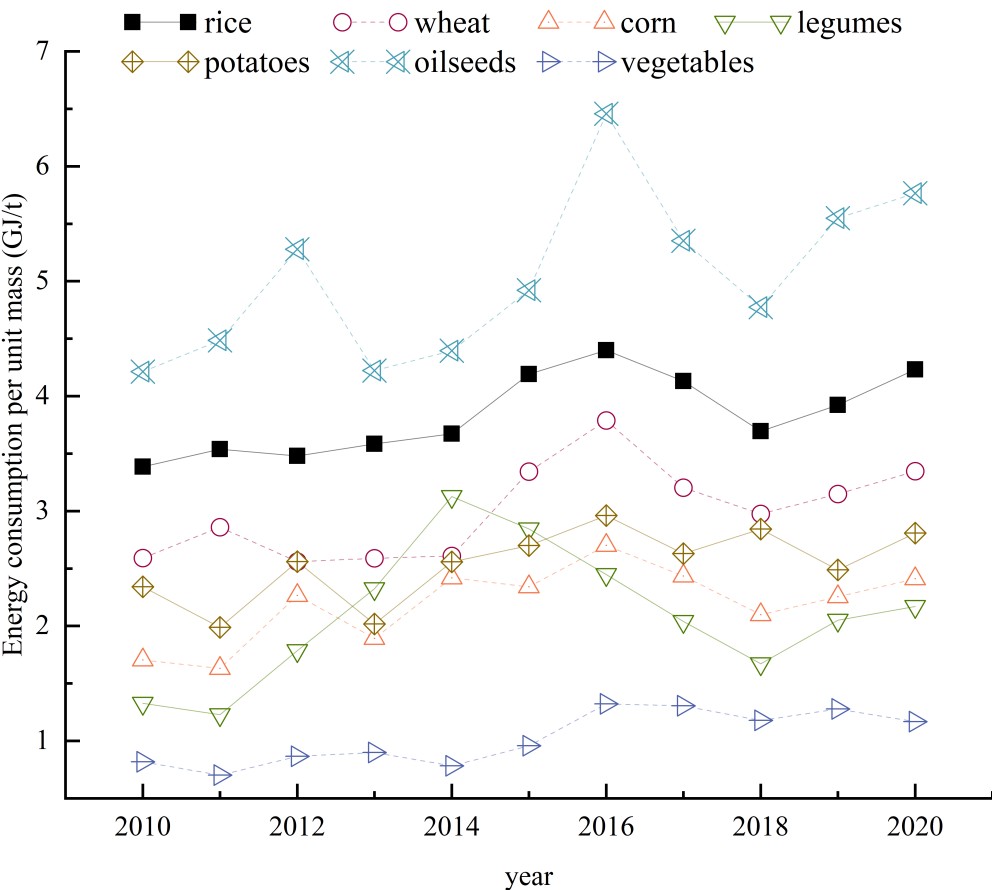

**Figure 6.** Annual variation in energy consumption per unit yield for crops.

*3.2. Jiangsu Province Agricultural Carbon Emission Temporal and Spatial Characteristics Analysis*

3.2.1. Temporal Characteristics

The total agricultural carbon emissions in Jiangsu Province showed a trend of initial increase followed by fluctuation and decline from 2010 to 2020. The year-on-year growth rate exhibited significant variations (Figure 7) and can be analyzed in two phases:

(1) 2010–2016: This was the ascending phase, with year-on-year growth rate exceeding 15% in 2015 and 2016. The main reason was the increased input of factors such as fertilizers and pesticides, intensifying the negative externalities of crop cultivation on the environment. During the "Twelfth Five-Year Plan" period, Jiangsu Province aimed to enhance the comprehensive production capacity of the agricultural industry and accelerate the modernization of agriculture. To meet the demand for increased grain production, the use of agricultural inputs continued to rise, leading to a continuous increase in carbon emissions.

(2) 2016–2020: This phase witnessed fluctuation and decline, with the most significant decrease occurring in 2017. The total carbon emissions in 2020 were $7.46 * 10^6$ *t C*, a 3.7% decrease compared to 2019. With the increasing severity of the greenhouse effect, environmental protection, previously overlooked due to economic growth, became a key focus area for sustainable development. Agricultural sustainable development necessitates reducing ecological damage and lowering greenhouse gas emissions while ensuring food security. In 2015, Jiangsu Province issued a notice on the "Zero Growth Action Plan for Fertilizer Use in Jiangsu Province by 2020." In 2017, the "Thirteenth Five-Year Plan" for modern agricultural development identified agricultural sustainable development as a major action, proposing the establishment of a comprehensive scientific fertilization management and technical system to effectively reduce chemical

fertilizer use in crop cultivation. The plan proposes to establish a comprehensive and scientific fertilization management and technical system, effectively reducing the application of chemical fertilizers in crop cultivation.

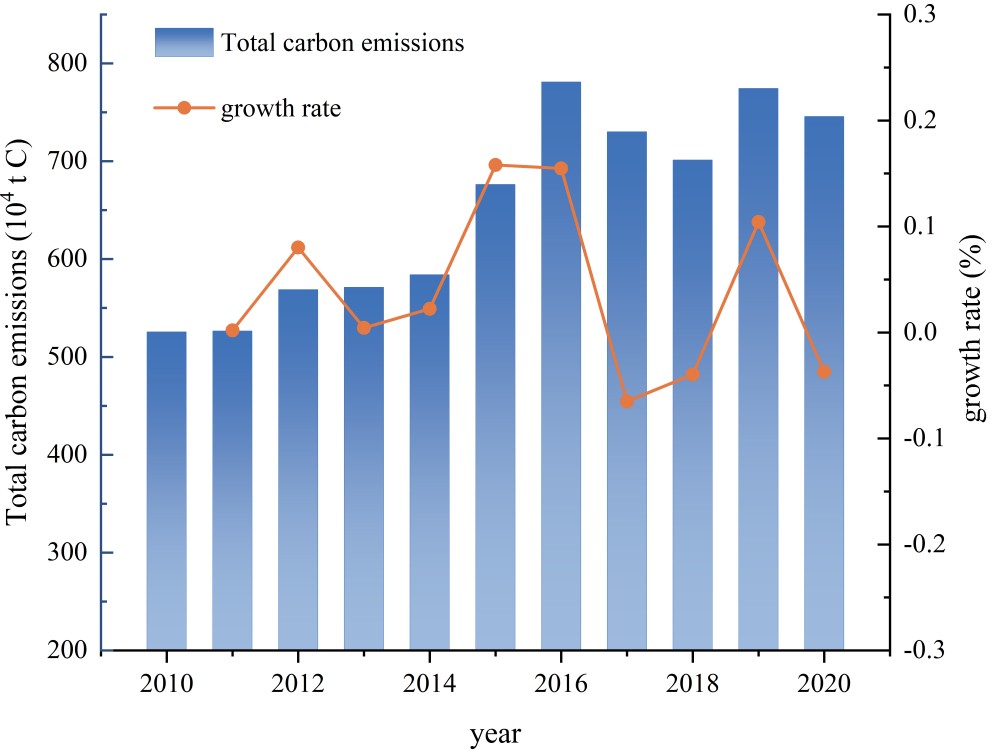

**Figure 7.** Total agricultural carbon emissions and year-on-year growth rate from 2010 to 2020.

The composition of agricultural carbon emissions and the results of the carbon emission intensity calculations are detailed in Table 2. It can be observed that the total carbon emissions induced by fertilizer input were the highest during the period from 2010 to 2020, accounting for 45% of the total emissions, with relatively small annual variations. Although the carbon emission coefficient of fertilizers is relatively small, they are crucial elements for crop yield increase and agricultural development. Therefore, until there is a significant breakthrough in crop production technology, farmers often enhance food supply capacity by increasing fertilizer usage.

The carbon emissions resulting from pesticide and plastic film usage account for 20% and 19%, respectively, of the total emissions. The carbon emissions from pesticides exhibit significant fluctuations and an overall upward trend, while those from agricultural films show a relatively small variation, slowly increasing until 2019 and decreasing in 2020. The proportion of carbon emissions caused by soil $N_2O$ is the smallest at 16%, and maintains a slow upward trend correlating with changes in the area of crop cultivation.

Carbon emission intensity is measured as the carbon emissions per unit area of crops. From 2010 to 2020, the carbon emission intensity exhibited a similar trend to the total carbon emissions. It increased before 2016, with an average annual growth rate of 4.32%, followed by a fluctuating downward trend. This indicates that effective results have been achieved in the control of agricultural carbon emissions with the promotion of green agricultural concepts and the dissemination of low-carbon technologies.

**Table 2.** Composition of agricultural carbon emissions and carbon emission intensity.

| | Carbon Emission Quantity ($10^4$ t C) | | | | Carbon Emission Intensity (kg C/hm$^2$) |
|---|---|---|---|---|---|
| | Rice | Wheat | Corn | Legumes | |
| 2010 | 268.3 | 79.0 | 82.4 | 95.9 | 750.57 |
| 2011 | 260.1 | 81.9 | 87.4 | 97.2 | 754.34 |
| 2012 | 265.1 | 85.8 | 118.3 | 99.6 | 768.79 |
| 2013 | 272.2 | 86.7 | 111.4 | 101.0 | 738.69 |
| 2014 | 280.2 | 86.4 | 115.6 | 101.8 | 773.89 |
| 2015 | 291.1 | 167.4 | 113.5 | 104.3 | 829.88 |
| 2016 | 331.8 | 205.5 | 139.7 | 104.0 | 957.96 |
| 2017 | 325.2 | 166.6 | 135.1 | 103.3 | 909.90 |
| 2018 | 333.8 | 125.4 | 134.9 | 107.2 | 879.23 |
| 2019 | 314.4 | 184.4 | 169.6 | 106.1 | 965.63 |
| 2020 | 309.6 | 195.5 | 133.9 | 106.8 | 933.72 |

### 3.2.2. Spatial Characteristics

We drew a spatial–temporal distribution map of agricultural carbon emissions and analyzed the spatial differentiation characteristics of agricultural carbon emissions in 2010, 2016, and 2020 (Figure 8). Zhenjiang, Changzhou, and Wuxi consistently have low levels of agricultural carbon emissions, while Xuzhou and Yancheng consistently exhibit high levels. Based on geographical space, Jiangsu Province is divided into three major regions: Southern Jiangsu (SuNan), Central Jiangsu (SuZhong), and Northern Jiangsu (SuBei). The total agricultural carbon emissions show a distinct pattern of SuBei > SuZhong > SuNan. In 2016, the entire SuBei region was in the high carbon emission zone; by 2020, however, the carbon emissions in Lianyungang City had significantly decreased. The carbon emissions of the three cities in SuZhong are moderate and have increased over time. Specifically, Yangzhou and Taizhou, which started at a low level in 2010, rose to a relatively low level in 2016 and 2020. Most areas in SuNan have consistently remained in the low carbon emission zone, with Nanjing consistently at a relatively low level and Suzhou rising from a low level in 2010 to a relatively low level. SuNan has a smaller crop cultivation area compared to other regions, indicating a higher level of green development with significant achievements in both technological development and environmental protection. SuZhong has a moderate level of agricultural development, with relatively less cultivation of rice and wheat. SuBei is an important grain-producing region, with large-scale cultivation of major crops. As agricultural modernization advances, agricultural carbon emissions have become an issue that requires attention.

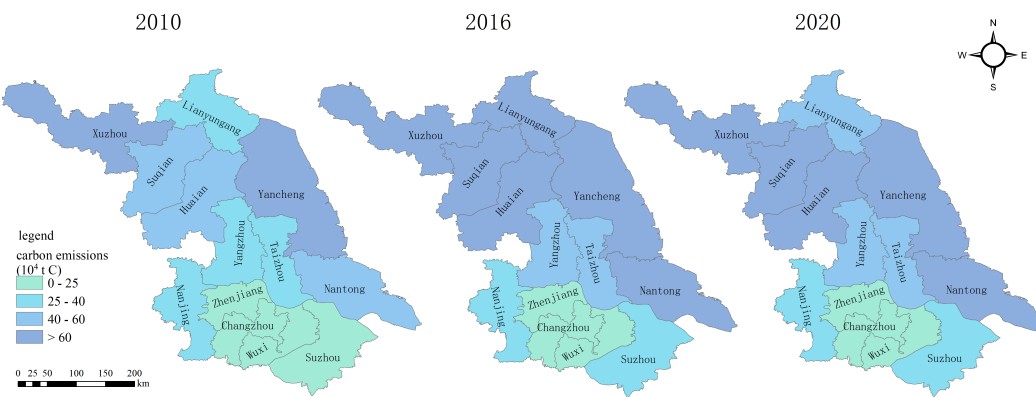

**Figure 8.** Distribution of agricultural carbon emissions in Jiangsu province in 2010, 2016, and 2020.

A spatial autocorrelation study was conducted on the carbon emission levels of various regions to further analyze their spatial clustering. The results of the global Moran's I index

calculation are presented in Table 3 and the Moran scatter plots for 2010, 2016, and 2020 are shown in Figure 9.

From 2010 to 2020, the agricultural carbon emissions passed the significance test and the Moran's I values were positive, indicating the presence of positive spatial autocorrelation in agricultural carbon emissions. The agricultural carbon emission situations of various cities are interconnected and mutually influential. Overall, the Moran's I values fluctuated, initially increasing, then decreasing, and increasing again with changes in the years. This suggests that while the degree of mutual influence arising from geographical distribution varies, there remains a noticeable tendency for convergence in development among adjacent regions.

**Table 3.** Global Moran index.

| Years | Moran's I | Z | P |
|---|---|---|---|
| 2010 | 0.090 | 3.201 | 0.001 |
| 2011 | 0.095 | 3.273 | 0.001 |
| 2012 | 0.092 | 3.242 | 0.001 |
| 2013 | 0.096 | 3.305 | 0.001 |
| 2014 | 0.096 | 3.303 | 0.001 |
| 2015 | 0.096 | 3.301 | 0.001 |
| 2016 | 0.087 | 3.163 | 0.002 |
| 2017 | 0.090 | 3.203 | 0.001 |
| 2018 | 0.091 | 3.238 | 0.001 |
| 2019 | 0.093 | 3.260 | 0.001 |
| 2020 | 0.103 | 3.412 | 0.001 |

From Figure 9, it is evident that the distribution pattern of the 13 cities in Jiangsu Province remains relatively stable across the four quadrants. Xuzhou, Yancheng, Suqian, and Huai'an are located in the HH (High-High Aggregation) region. These cities are situated in the primary agricultural production area of northern Jiangsu and share similar planting structures, with generally higher agricultural carbon emissions that have a significant impact on the surrounding cities. Yangzhou and Taizhou are in the LH (Low-High Aggregation) region. These two cities are in the transitional zone of central Jiangsu, exhibiting lower carbon emissions themselves while being adjacent to the higher agricultural carbon emissions in northern Jiangsu. Lianyungang moved from the LH region to the HH region, indicating that its agricultural carbon emissions, influenced by surrounding cities, increased from a lower level to a higher level.

Nanjing, Suzhou, Zhenjiang, Wuxi, and Changzhou are in the LL (Low-Low Aggregation) region. These cities belong to the more developed south of Jiangsu, focusing on the cultivation of rice, wheat, and vegetables. Their overall agricultural carbon emissions are relatively low, and due to their similar development status they exhibit positive effects in neighboring spatial areas. Only Nantong is in the HL (High-Low Aggregation) region. This city, located adjacent to Suzhou in the south, Yancheng in the north, and Taizhou in the west, is an important oilseed crop production area with relatively higher agricultural carbon emissions.

In light of the imbalance in the development of Jiangsu Province, the development plan of "Promoting South Jiangsu, Rising Central Jiangsu, and Revitalizing North Jiangsu" was proposed early in the new century. In the current context of climate change, it is essential to focus on the green and coordinated development of agriculture.

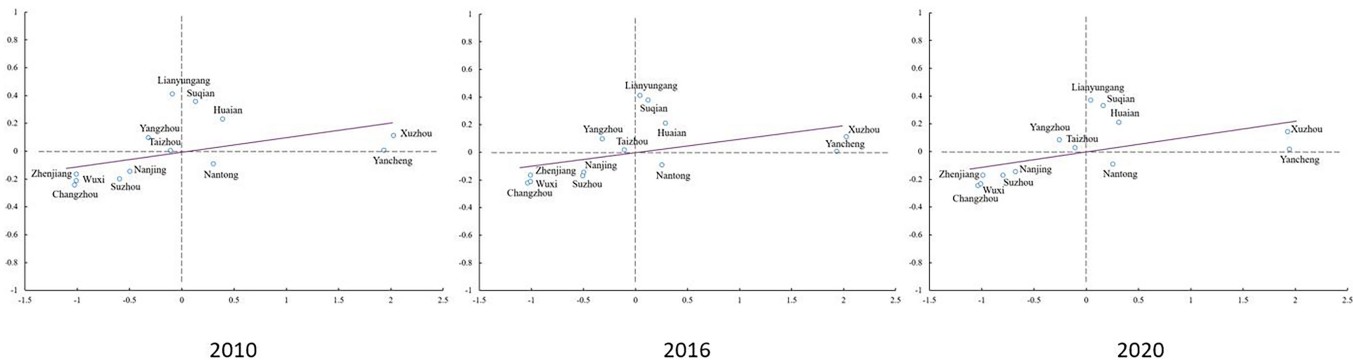

**Figure 9.** Local Moran scatter plot of carbon emissions in 2010, 2016, and 2020.

*3.3. Model Solving and Optimization Results of Planting Structure*

This paper employs genetic algorithms to solve the multi-objective optimization model. A normalization method is adopted to preprocess the objective functions while eliminating the influence of dimensional differences. The optimization results are presented in Table 4.

**Table 4.** Results of crop planting structure optimization.

| | 2020 | | Optimized | | |
|---|---|---|---|---|---|
| | Planted Area ($10^4$ hm$^2$) | Proportion | Planted Area ($10^4$ hm$^2$) | Proportion | Proportional Change |
| Rice | 220.284 | 31.45% | 263.116 | 36.68% | 5.25% |
| Wheat | 233.889 | 33.40% | 112.215 | 15.64% | −17.73% |
| Corn | 50.976 | 7.28% | 61.992 | 8.64% | 1.37% |
| Legumes | 19.639 | 2.80% | 92.265 | 12.86% | 10.06% |
| Potatoes | 3.907 | 0.56% | 19.425 | 2.71% | 2.15% |
| Oilseeds | 27.261 | 3.89% | 29.095 | 4.06% | 0.08% |
| Vegetables | 144.38 | 20.62% | 139.318 | 19.42% | −1.18% |
| Total | 700.336 | 100% | 717.426 | 100% | — |

From the results in Table 4, it can be observed that the planting area and proportion of crops in Jiangsu Province have changed after optimization. The total planting area has increased from $7.00 * 10^6$ hm$^2$ to $7.17 * 10^6$ hm$^2$, with a growth rate of 2.44%. Wheat, initially the crop with the largest planting area, accounted for 33.37% of the total, while the planting area of potatoes was the smallest at only $3.91 * 10^5$ hm$^2$, accounting for less than 1%. After adjustment, the proportion of legumes has increased the most, rising by 10.06 percentage points. This is because legumes have relatively high economic benefits along with relatively small carbon emissions and grey water footprints. Thus, increasing the planting area can yield relatively high comprehensive benefits. The proportion of wheat planting has decreased the most, by 17.73%. This is due to the relatively low economic benefits of wheat. Rice and vegetables, as important crop varieties, play a significant role in meeting people's daily needs. Minimum planting areas have been set for them in development plans. The proportion of rice planting has increased by 5.25%, while the proportion of vegetable planting has decreased by 1.18%. Both total planting areas meet the area thresholds while ensuring adequate market supply. Additionally, the proportions of potatoes, corn, and oilseed planting have all increased slightly, by 2.15%, 1.37%, and 0.08%, respectively. Analysis reveals that while these are high-profit crops, they cause higher pollution. In the future, introducing superior varieties and exploring balanced crop development paths could be considered.

After optimization of the planting structure, various objectives of Jiangsu Province's planting industry have shown improvement(Table 5). Economic benefits have increased from $1.632 * 10^{11}$ CNY to $1.689 * 10^{11}$ CNY, with a growth of $6.63 * 10^9$ CNY, representing a 4.06% increase. Carbon emissions have decreased by $2.839 * 10^5$ t C, from $7.503 * 10^6$ t C of

carbon to $7.220 * 10^6$ t C, a reduction of 3.78%. The total grey water footprint has decreased from $1.056 * 10^{10}$ m$^3$ to $9.758 * 10^9$ m$^3$, a decrease of 7.61%. The economic benefits per unit water footprint have increased by 0.17 CNY/m$^3$, and the economic benefits per unit energy consumption have increased by 36.21 CNY/GJ. Overall, the optimized planting structure shows improvements in both economic and ecological benefits, meeting the requirements of the "Comprehensive Promotion of Rural Revitalization and Accelerating Agricultural Modernization in Jiangsu Province during the 14th Five-Year Plan". It shows reduced carbon emissions and grey water footprints while controlling crop water resources and energy consumption, thereby promoting the green and sustainable development of the planting industry.

**Table 5.** Comparison of elements before and after optimization.

| | Benefits (10$^8$ CNY) | Carbon Emission (10$^4$ t C) | Grey Water Footprint (10$^8$ m$^3$) | Water Footprint (10$^8$ m$^3$) | Energy Consumption (10$^4$ GJ) |
|---|---|---|---|---|---|
| 2020 | 1631.80 | 750.34 | 105.63 | 493.64 | 20,258.40 |
| Optimized | 1698.12 | 721.95 | 97.58 | 488.64 | 20,174.73 |

## 4. Discussion

The results of water footprint calculations for three crucial cereal crops, namely, rice, wheat, and maize, were compared with those of Xu et al. [65]. While our computed water footprint values are slightly higher than those obtained by Xu et al. for the same three crops in 2019, the relative size relationships are generally consistent. This is attributed to the inclusion of grey water footprint in our study and discrepancies in the processing of meteorological data.

Jiang et al. [66] pointed out that diesel is the primary source of energy consumption in the agricultural sector. The research in this paper indicates that among the energy consumption of five grain crops, the proportion of diesel exceeds 18% in all cases, with diesel accounting for as much as 45.67% of the energy consumption in legumes.

Hu et al. [52] conducted a spatial distribution study on agricultural carbon emissions in Jiangsu Province, revealing that high-emission areas are concentrated in northern Jiangsu, displaying strong spatial agglomeration characteristics. Despite variations in measurement scope, their conclusions closely align with the findings of this paper. The potential for carbon emission reduction in Jiangsu Province's agriculture sector requires further exploration.

The current grain production in Jiangsu Province can generally meet the needs of local residents [67]. While safeguarding the red line of arable land and ensuring grain production are priorities, enhancing economic benefits and reducing environmental pollution are of utmost importance. The setting of parameters in multi-objective planning should fully consider optimization goals and practical circumstances. In this paper, model parameters were established based on relevant policy plans and calculation results, following the approach outlined in reference [68].

Overall, the optimized planting areas of the seven main crops have seen a slight increase. Studies have shown that there is a negative correlation between planting structure and farmland pressure [69]. Due to the relatively high farmland pressure in Jiangsu Province [70] and the high rate of non-grainification [71], appropriately reducing the planting area of some non-grain crops can alleviate farmland pressure. Among these, the planting area of vegetable crops has decreased slightly in the optimized results presented in this paper, while the planting area of other grain crops (excluding wheat) has increased. The proportion of grain crops in the total sown area has increased by 1.10 percentage points, which is a more reasonable adjustment direction.

The increase in the planting area of legumes is the most significant. China's heavy reliance on imported soybeans is prominent, with relatively low comparative benefits and

low willingness among farmers to plant them [72]. Thus, it is necessary to take measures to encourage the cultivation of legumes. The planting area of the main grain crop, wheat, has decreased. Wheat cultivation is widespread in China, with the main cultivation areas concentrated in the northern regions [73], particularly in Henan, which is the largest wheat-producing province. Therefore, although the wheat planting area in Jiangsu Province has been reduced in the optimized results, the shortfall can be accommodated by the production areas in the north. There has been some increase in the planting area of corn. Local corn production in Jiangsu Province is relatively low, with long-term reliance on imports, resulting in a significant gap between production and demand [74]. The optimized results are conducive to reducing the circulation pressure caused by the imbalance between production and demand. Potatoes had the smallest proportion in the initial planting structure; however, the planting area has seen a certain increase after optimization. This can strengthen the experimentation and cultivation of excellent varieties and broaden the industrial layout of the potato industry.

Overall, this paper provides an accurate analysis of resource consumption and pollution in the process of crop production which is consistent with the local reality. Furthermore, the optimization results of the planting structure align with the requirements for green and sustainable development as outlined in documents such as the "Implementation Opinions on Accelerating the Promotion of Green Agricultural Development" and the "Action Plan for Building a Strong Province in Agriculture" issued by Jiangsu Province. The adjustments made based on this analysis demonstrate a degree of feasibility in practical implementation.

Meanwhile, exploring the research results and their impact on policy-making and related industries can facilitate the practical implementation of the plans. According to scholars' studies, while the agricultural infrastructure construction in Jiangsu Province is basically sound, there are some issues around anti-intensive land use [75]. This is reflected in phenomena such as the reduction of planting areas and the marginalization of arable land. Providing macroscopic guidance and planning for crop planting can improve the efficiency of agricultural output and help move towards sustainable intensification. The results of this paper's analysis show that energy consumption and carbon emissions of major crops in Jiangsu Province both show fluctuating upward trends, with the possibility of continued growth. Low-carbon agricultural technologies, such as deep plowing, straw cover, and fertilizer control, can reduce carbon emissions [76]. It is imperative to introduce appropriate low-carbon agricultural technologies and establish corresponding management systems.

For farmers engaged in specific crop planting activities, several factors influence their choice of crop types. Past planting experience, support policies in the current year, and potential economic returns all play a role in their decision-making. Among these factors, agricultural subsidy policies have always been a major policy measure in the field of agricultural development and require adjustments in alignment with social and economic development [77]. Appropriately increasing subsidy efforts can enhance their guiding role in the development of the planting industry.

Based on the above discussion and analysis, the following recommendations are proposed:

(1) Enhance agricultural planting planning. Jiangsu Province faces relative scarcity of arable land resources. While strictly adhering to the constraints on arable land area, it is necessary to fully leverage the resource advantages of various regions while orienting towards market demand. This involves rational planning and management of key crop production areas, scientifically laying out agricultural supporting facilities, providing policy support, nurturing, selecting, and improving high-yield high-quality crop varieties, and emphasizing precise harvesting and processing of crop products to ensure both quantity and quality.

(2) Regulate production water and energy inputs. Utilize the latest technological means to accurately monitor water resource consumption and the utilization efficiency of agricultural materials throughout the crop production process. This will help alleviate the uneven distribution of resources caused by large-scale planting. Providing

advanced technological assistance to production parks. From selecting seeds and planting to field irrigation, pesticide and fertilizer application, timely irrigation cessation, drainage, and post-harvest storage, optimize each step gradually to reduce water and energy wastage during the planting process.

(3) Promote low-carbon farming practices. Introduce technologies such as soil testing and formula fertilization, integrated water and fertilizer management to enhance the efficiency of fertilizer and pesticide usage. Additionally, implement recycling programs for agricultural waste to reduce soil pollution from residual substances. Collaborate with relevant national agencies to accurately account for agricultural greenhouse gas emissions, considering both direct emissions during production and indirect emissions throughout the product lifecycle. Learn from the experience of green agriculture development in southern Jiangsu, address shortcomings, and establish government-led low-carbon agricultural production systems.

(4) Increase agricultural subsidy efforts. According to relevant laws and regulations on agricultural financial subsidies, determine the levels and boundaries of subsidies that farmers can receive. For crops encouraged to be planted such as legumes and corn, the amount of subsidies should be increased. This is to offset some of the costs incurred by transitioning production methods. At the same time, improve the efficiency of subsidy implementation, keep detailed records of each step, make the entire process transparent, and ensure the interests of all relevant parties.

## 5. Conclusions

This article calculates the water footprint, energy consumption, and carbon emissions of seven crops in Jiangsu Province. It optimizes solutions based on a multi-objective planning model of crop structure. The main conclusions are as follows:

(1) From 2010 to 2020, the total water footprint of crops fluctuated significantly, showing a downward trend in recent years. Blue water and green water have similar proportions, with the green water footprint of rice slightly higher than the blue water footprint. The grey water footprint of vegetables is relatively high. Rice, with the largest planting area, has a high unit area water footprint and a moderate unit yield water footprint. Despite the small planting area, beans and oilseeds have higher unit yield water footprints.

(2) From 2010 to 2020, the total energy consumption of crops showed a fluctuating upward trend, with rice contributing the highest total energy consumption. Looking at the energy consumption structure per unit area of crops, diesel and fertilizers are the main sources of energy consumption, and rice and vegetables have a significant demand for irrigation electricity. Oilseeds have the highest unit yield energy consumption, while vegetables have the lowest. The average energy consumption of other crops is between 2 and 4 GJ/t.

(3) The total agricultural carbon emissions showed a trend of first increasing and then fluctuating and decreasing between 2010 and 2020. The year-on-year growth rate exhibited noticeable variations, with the most significant decrease observed in 2017. The carbon emissions from chemical fertilizers, pesticides, and plastic films accounted for 45%, 20%, and 19%, respectively. The carbon emissions from soil $N_2O$ accounted for 16%. Agricultural carbon emission intensity showed a similar trend to total carbon emissions, with an average annual growth rate of 4.32% before 2016 and a fluctuating decline thereafter.

(4) After optimizing the planting structure, there was a significant increase in the proportion of legumes, while the proportion of wheat decreased noticeably. Other crops with adjustable area proportions included rice, potatoes, corn, and oilseeds, in descending order. The proportion of vegetable cultivation slightly decreased, and the output of important crops met the demand, aligning with the policy planning for the development of dry grain crops. The optimized plan resulted in a 4.06% increase in

economic benefits, along with a 3.78% and 7.61% reduction in carbon emissions and greywater footprint, respectively, indicating a favorable optimization outcome.

(5) Reasonably promoting optimized planting structure schemes can enhance the overall efficiency of crop production. Policymakers should work on strengthening agricultural planting planning, regulating the input of water and energy in production, promoting low-carbon planting models, and increasing agricultural subsidy efforts. By better leveraging local resource advantages, it is possible to achieve green and sustainable development of the planting industry while meeting demand.

In future research, we plan to explore additional socioeconomic factors that may influence planting structure and assess the applicability and scalability of the model in other regions. Methodologically, we will integrate geographic information system (GIS) and other spatial technologies to examine the impact of topography and land quality variations on crop planting.

**Author Contributions:** Y.J.: conceptualization, methodology, data curation, writing—original draft, writing-reviewing and editing, visualization; X.Y.: conceptualization, methodology, writing—reviewing and editing. All authors have read and agreed to the published version of the manuscript.

**Funding:** This work was financially supported by the Major Project of National Social Science Foundation of China (Grant No. 19ZDA084) and the Fundamental Research Funds for the Central Universities (Grant No. B230207002).

**Institutional Review Board Statement:** Not applicable.

**Informed Consent Statement:** Not applicable.

**Data Availability Statement:** Data will be accessible from the author group if requested.

**Acknowledgments:** We appreciate all the researchers who provided information and assistance for this study.

**Conflicts of Interest:** The authors declare no conflicts of interest.

## Abbreviations

| | |
|---|---|
| SDGs | Sustainable Development Goals |
| FAO | Food and Agriculture Organization of the United Nations |
| SuNan | Southern Jiangsu |
| SuZhong | Central Jiangsu |
| SuBei | Northern Jiangsu |
| HH | High–High Aggregation |
| LH | Low–High Aggregation |
| LL | Low–Low Aggregation |
| HL | High–Low Aggregation |
| GIS | Geographic Information System |

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
