# Peer review of "Multi-Objective Optimization of the Planting Industry in Jiangsu Province and Analysis of Its “Water-Energy-Carbon” Characteristics"

_sustainability, doi:10.3390/su16072792_

Round 1

Reviewer 1 Report

Comments and Suggestions for Authors

The paper is well written and sufficiently clear. I have no particular remarks; just suggest authors to improve the dicussion section with some more considerations on the practical feasibility of the optimal scenario and its possible implications for farmers and the agrofood industries.

Comments on the Quality of English Language

The paper is well written and makes use of good syntax and grammar

Reviewer 2 Report

Comments and Suggestions for Authors
  1. Clarity and Contextualization: The research effectively outlines the importance of modern agricultural development in addressing both food security and environmental concerns. However, further contextualization regarding the specific social and environmental issues faced by Jiangsu Province could enhance the reader's understanding of the research context.

  2. Methodology and Analysis:

    • The incorporation of water footprint, energy consumption, and carbon emissions analysis provides a comprehensive view of the environmental impact of crop cultivation.
    • The establishment of a multi-objective planning model demonstrates a structured approach towards optimizing planting structures.
    • However, additional information on the methodologies employed for data collection, analysis, and model establishment would enhance the transparency and reproducibility of the study.
  3. Results and Findings:

    • The presentation of trends in water footprint, energy consumption, and carbon emissions over the study period offers valuable insights into the dynamics of agricultural sustainability.
    • Spatial analysis of carbon emissions adds depth to the findings, highlighting regional disparities and clustering characteristics.
    • Quantitative results regarding the optimization of planting structures, such as the increase in economic benefits and reduction in carbon emissions and water footprints, are significant. However, providing insights into the specific interventions or changes in cropping patterns contributing to these improvements would enrich the discussion.
  4. Discussion and Implications:

    • The discussion on the implications of the findings for adjusting planting structures and promoting green and sustainable development is commendable.
    • Elaborating on the potential policy implications or practical recommendations arising from the study's results would strengthen its relevance and utility for stakeholders involved in agricultural planning and management.
  5. Future Directions:

    • Suggestions for future research avenues, such as exploring the socio-economic factors influencing planting structure optimization or assessing the scalability of the proposed model to different regions, could be beneficial for advancing knowledge in this field.

Overall, the research contributes valuable insights into the environmental sustainability of crop cultivation in Jiangsu Province and provides a foundation for further investigation into optimizing planting structures for green and sustainable development. Strengthening the methodological transparency, contextualization, and discussion of implications would enhance the research's impact and relevance.

Comments on the Quality of English Language

Revisions re required for exact use of grammar.

Reviewer 3 Report

Comments and Suggestions for Authors

I see that the paper has been checked and the Authors made improvements. However, some improvements can still be added.

1. The aim in introduction and abstract should be the same.

2. The introduction section should point out the research gap. The attention should be focused on what are the new main issues.

3. The paper does not contain the literature review chapter, so at the end of intruduction the authors of the paper can elaborate hypothese and write short paragraph how the paper is organised.

4. In my opinion the methods are well described however it takes 4 pages what seems too long.  

5. In my opinion the topic is very interesting both for readers and for policy makers. That is why the conclusion section should include some implications for policy.

6. Most of the cited references are from well recognized journals, however the authors forgot to write pages at the end, for example references number: 1,5, 6, 11, 13, 17, 20, 21, 22, 24, 25, 26, 35, 37, 38, 39, 40, 41, 42, 43, 44, 45, 46, 47, 48, 49, 50, 51, 53, 54, 56, 61, 62, 65, 66, 67. Please complete pages.

7. The paper should contain nomenclature (abbreviations) at the end of the paper before references. 

8. From scientific side the paper is elaborated quite well, however figure 1 overall approach to research is difficult to understand. Please make is easier to understand. 

9. The authors presented 22 equations what is quite a large number. In my opinion it is better to focuse on results description rather that coping equations.

10. After correcting the paper can be accepted for publication.

Comments on the Quality of English Language

English is good.

Round 2

Reviewer 2 Report

Comments and Suggestions for Authors

Improvements are satisfactory.

Comments on the Quality of English Language

Acceptable